# COVID-19 Vaccination among Czech Dentists

**DOI:** 10.3390/vaccines10030428

**Published:** 2022-03-11

**Authors:** Jan Schmidt, Vojtech Perina, Jana Treglerova, Nela Pilbauerova, Jakub Suchanek, Roman Smucler

**Affiliations:** 1Department of Dentistry, Faculty of Medicine in Hradec Kralove, Charles University and University Hospital Hradec Kralove, 500 05 Hradec Kralove, Czech Republic; jan.schmidt@lfhk.cuni.cz (J.S.); nela.pilbauerova@lfhk.cuni.cz (N.P.); suchanekJ@lfhk.cuni.cz (J.S.); 2Department of Oral and Maxillofacial Surgery, Faculty of Medicine, Masaryk University and University Hospital Brno, 625 00 Brno, Czech Republic; treglerova.jana@fnbrno.cz; 3Department of Stomatology, 1st Faculty of Medicine, Charles University and General Teaching Hospital, Katerinska 32, 121 08 Prague, Czech Republic; smucler@dent.cz

**Keywords:** COVID-19, SARS-CoV-2, vaccination, dentistry, pandemic, dentist, occupational health

## Abstract

This work describes and evaluates vaccination against COVID-19 among members of the Czech Dental Chamber during the pre-booster vaccination phase. A cross-sectional online survey was conducted between 23 June and 4 September 2021, among 2716 participants, representing 24.3% of all chamber members. A total of 89.5% of respondents stated that they were registered for vaccination against COVID-19, their vaccination had started or been completed, or had a medically relevant reason to avoid vaccination. A total of 79.6% of respondents stated that they were fully vaccinated, most of them with the Comirnaty (Pfizer–BioNTech) vaccine (88.3%). The vaccination rate among males was significantly higher than among females (*p* = 0.001, OR 1.48). The main reasons for vaccination were professional (91.5%). The share of fully vaccinated participants was significantly higher (*p* < 0.0001, OR = 8.17) compared to the Czech general population (30.8%). A COVID-19 vaccine breakthrough infection rate was 0.42%. The study shows that both the willingness to vaccinate and the proportion of fully vaccinated individuals among Czech dentists are high, and only about 10% of them refused vaccination based on reasons classified as not medically relevant.

## 1. Introduction

In the first half of 2021, the Czech Republic was the fourth most affected country by COVID-19 in the world [1,2,3]. Public health restrictions caused by the COVID-19 pandemic severely affected the entire country, including the healthcare system. The rapid increase in the number of COVID-19 patients resulted in a suspension of non-acute general healthcare in the Czech Republic. Such an approach was similar in most countries of the European Union. Despite general healthcare suppression, one of our previous studies revealed that Czech dentists worked even throughout the pandemic, and Czech dentistry remained more than 90% operative throughout the COVID-19 pandemic [4]. Such an approach was rare on a European and global scale as dentists are one of the COVID-19 highly vulnerable professional groups [4,5]. These factors—the high-level prevalence of COVID-19 in the Czech Republic, the high workload during the pandemic, and the significant risk of infection due to dentists’ work settings—make Czech dentists a unique epidemiological group.

Based on this, the Czech Dental Chamber decided to conduct an extensive survey to determine the impact of COVID-19 on its members. The first phase of the research contained several chapters of different focuses (see Materials and Methods). The chapter focused on COVID-19 prevalence among Czech dentists revealed that despite the high occupational risk, the prevalence among dentists was significantly lower than among the Czech general population [3]. The study indicated that although the dental profession is associated with a high occupational risk of droplet infection transmission, including SARS-CoV-19, the working conditions of dentists in the Czech Republic have not led to a higher prevalence of COVID-19 infection among them. This could be due to well-chosen anti-epidemic measures adopted by Czech dentists. The results of the next chapter focused on vaccination are presented in this work. The survey is part of a long-term study assessing the impact of COVID-19 on Czech dentists, with the next phase to be initiated in 2022.

The impact of COVID-19 on dentists is a topic that is not sufficiently described. Several works covered the pandemic’s impact on the operation of dental practices, but only a few studies focused on the impact on dental professionals [6,7,8,9,10]. Our team was the first to publish a study on the national COVID-19 prevalence of dentists from the beginning of the pandemic to the end of the first half of 2021 [3]. Although dentists are one of the most vulnerable occupational groups, information about COVID-19 vaccination among them is scarce. There is only a very limited amount of works on this topic, and in the vast majority, it only concerns the willingness, hesitancy, or acceptance of dentists to get vaccinated [11,12,13,14,15]. To the best of our knowledge, this is the first study to provide data on real COVID-19 vaccination among dental professionals.

## 2. Materials and Methods

### 2.1. Design

The Czech Dental Chamber designed this questionnaire in collaboration with the academic community, experts from the chamber, and general practitioners. The resulting questionnaire was sent together with the request to participate in this self-administered, cross-sectional survey to the officially registered e-mail addresses of the chamber members. The e-mail also included information on the purpose of the study. Participants’ responses were anonymous, untraceable, and did not contain any identifying personal information. No remuneration was provided to the respondents, and completing the questionnaire did not bring them any direct benefits. This study was conducted in accordance with the Declaration of Helsinki. The survey and the content of the questionnaire were approved and supervised by the Executive Board of the Czech Dental Chamber.

Since the entire survey was designed as a tool to widely analyze the impact of COVID-19 on the Czech Dental Chamber members in various aspects, it was divided into several chapters. Each chapter covered a different topic, namely the following: Prevalence, Vaccination, and Epidemiology. Due to data extension and the different focus of individual chapters, their results were analyzed and presented separately. The results of the Prevalence chapter were published in November 2021, the data of the Vaccination chapter are presented in this work, and the Epidemiology chapter will be analyzed in 2022 [3].

The presented data were obtained from the answers to 9 questions—4 questions were close-ended, and 5 were semi-close-ended (prefilled close-ended answers along with the option to reply in an open form). The questions and answers were exclusively in the Czech language throughout the questionnaire. The wording of the questions and their further specifications are available in Appendix A.

### 2.2. Sample

The Czech Dental Chamber sent an official invitation to participate in the survey to a total of 9922 officially registered e-mail addresses of its members. Each of these addresses was unique and belonged to an individual member. Participation in the questionnaire was possible from 23 June 2021 to 4 September 2021.

The latest official data showed that the chamber had 11,160 members as of 31 December 2020. This means that 88.9% of the members were addressed by this survey. The Czech Dental Chamber membership is compulsory for all dentists working in the country.

### 2.3. Sample Size Relevancy

The minimum number of research participants (*n*) to achieve relevant results was calculated as 372. The calculation was performed using Formula (1) via an online Netquest calculator. A study universe (*N*) was quantified as the number of the Czech Dental Chamber members (*n* = 11,162), a margin of error (*e*) was set at 5%, a confidence level (*Z*) at 95%, and a standard heterogeneity (*p*) at 50%. The statistical relevance of this study was confirmed as the number of participants (2716) exceeded the minimum required number.
(1)$$n=\frac{N\cdot Z^{2}\cdot p\cdot \left(1-p\right)}{\left(N-1\right)\cdot e^{2}+Z^{2}\cdot p\cdot \left(1-p\right)}$$

Formula (1). Relevant sample size calculation.

### 2.4. Data Collection

The e-mail sent as part of the invitation redirected the participants to an online questionnaire in Google Forms (Google, Mountain View, CA, USA). The questionnaire was compatible across the most commonly used display devices and operating systems. The response results were stored in the Google Forms cloud database during ongoing data collection. After data collection was completed, the results were downloaded.

### 2.5. Statistical Analysis

The data were downloaded from the Google Form cloud database and analyzed. Answers within close-ended questions are presented as a percentage of answers within all answers. The semi-close-ended question was analyzed independently by 3 authors (V.P., J.T., and J.S. (Jan Schmidt)). Results that did not agree between the authors were decided by the fourth author (J.S. (Jakub Suchanek)). The open responses were individually evaluated and grouped according to their meaning into predefined groups or a new group. A new group was created if the number of responses exceeded the threshold specified as *n* = 5. If this limit was not exceeded, the responses were assigned to the “Others” group. The results are presented as a percentage of answers within all answers. Empty answers were not counted.

A person was considered partly vaccinated if they have received only one dose of a 2-dose vaccine protocol. A person was considered fully vaccinated if they have received a single-dose vaccine or both doses of a two-dose vaccine.

The data were organized and analyzed in Microsoft Office Excel (version 2106 for Windows, Microsoft Corporation, Redmond, WA, USA) and GraphPad Prism (version 8.0.0 for Windows, GraphPad Software, San Diego, CA, USA).

## 3. Results

### 3.1. Response Rate

The questionnaire was filled out by 2716 out of 9922 addressed members of the chamber. The response rate was 27.4%. A total of 24.3% of all chamber members (*n* = 11,162) participated in the questionnaire [16].

### 3.2. Sex Distribution

Overall, sex information was provided by 2708 participants. A total of eight participants did not reply to this question. Females counted for 1871 (69.1%) respondents, while males for 837 (30.9%). This distribution corresponds to the overall distribution of females and males within the chamber population. In the study population, i.e., all members of the chamber (*n* = 11,160), there were 67.2% of females and 32.8% of males (illustrated in Figure 1).

### 3.3. Age Distribution

The question on age was answered by 2712 accountants, and four left it unanswered. The median age group of the survey participants is 50–60 years. The age of the study participants and the study population, i.e., all members of the chamber, is shown in Figure 1. Compared to the study population, fewer people under the age of 30 and over the age of 70 are represented among the study participants.

### 3.4. COVID-19 Vaccination

#### 3.4.1. Approach to COVID-19 Vaccination in the Entire Study Population

This question was answered by 2703 respondents; 13 respondents skipped this question. A total of 2243 respondents (83.0%) stated that their vaccination has started or is completed, 436 (16.1%) stated that their vaccination has neither started nor been completed, and 24 (0.9%) stated that they are registered for vaccination. Groups waiting for vaccination, with started or completed vaccination, account for 83.9% (illustrated in Figure 2).

#### 3.4.2. Reasons Leading to COVID-19 Vaccination

Out of 2267 respondents who stated that they are registered for vaccination or their vaccination has started or been completed, 2260 stated a reason for their vaccination. A total of 2069 (91.5%) replied it was due to their profession, 96 (4.2%) were vaccinated due to their age, 49 (2.2%) due to their health condition, 25 (1.1%) wanted to avoid restriction for unvaccinated people, 10 (0.4%) stated some kind of inner motivation, and the reasons of 11 (0.5%) participants were classified as “Others”.

#### 3.4.3. Reasons Leading to Avoiding COVID-19 Vaccination

Out of the 436 respondents who have not been registered for vaccination or stated that their vaccination has neither been started nor been completed, 429 replied to a question about the reason for avoiding vaccination. A total of 215 (50.1%) stated that they do not want to be vaccinated, 78 (18.2%) replied that they have laboratory tested anti-COVID-19 antibodies, 42 (9.8%) reported medical contraindication, 26 (6.1%) were diagnosed with COVID-19 in the last 180 days, 19 (4.4%) were pregnant or breastfeeding, 43 (10%) did not want to answer, and 6 (1.4%) stated reasons classified as “Others,” which are Illustrated in Figure 3.

Based on this summary, 165 respondents from the group unregistered for vaccination and unvaccinated stated medically relevant reasons for avoiding vaccination (antibodies, COVID-19 infection in the last 180 days, medical contraindication, pregnancy, and breastfeeding). The reasons “I do not want to” and “I do not want to reply” were considered as medically irrelevant reasons for avoiding vaccination. Pregnancy and breastfeeding as medically relevant reasons for avoiding vaccination are further discussed in the Discussion. In summary, 89.5% of all participants were either registered for vaccination, partly or fully vaccinated, or had medical reasons to avoid vaccination.

#### 3.4.4. Share of Participants Vaccinated against COVID-19 in the Entire Study Population

Out of the total of 2716 participants, 2163 (79.6%) stated that they are fully vaccinated against COVID-19, and 86 (3.2%) stated that they are partly vaccinated against COVID-19 (illustrated in Figure 2). It is important to note that “The rest” category does include individuals who had medically relevant reasons to avoid vaccination. This methodology was adopted to maintain comparability with other COVID-19 vaccination statistics, including Czech national vaccination statistics.

#### 3.4.5. Share of Participants Vaccinated against COVID-19 Based on Sex

A total of 2708 participants stated their sex. The categories fully vaccinated/partly vaccinated/the rest for males were 84.1%/2.4%/13.5%, respectively, and for females the were 77.7%/3.5%/18.8%, respectively. The difference in vaccination (fully and partly vaccinated vs. the rest) between males and females was statistically significant (*p* = 0.001; chi-square with Yates’ correction) with an odds ratio (OR) of 1.48 (95% confidence interval (CI) 1.045 to 1.862), which is illustrated in Figure 4.

#### 3.4.6. Share of Participants Vaccinated against COVID-19 Based on Age

The categories fully vaccinated/partly vaccinated/the rest were further sorted based on age. The results are shown in Figure 5. A detailed analysis of each age group and sex is provided in Appendix A.

#### 3.4.7. Time Distribution of Achieving Full Vaccination

Respondents who stated that they had completed vaccination against COVID-19 were asked when their vaccination was completed. Out of them, 2160 participants replied to this question. None of the respondents indicated that their vaccination was completed before December 2020 or after June 2021. The detailed results are provided in Appendix A.

#### 3.4.8. Comparison of Full COVID-19 Vaccination within the Study Population and the General Population in the Czech Republic

The share of fully vaccinated participants in this study was 79.6%. As none of the respondents indicated that their vaccination was completed after June 2021, the end of the reference period for this comparison was set for 30 June 2021. Thus, this chapter compares the share of fully vaccinated persons within this study population and the general Czech population as of 30 June 2021. According to data, as of 30 June 2021, 30.8% of the population in the Czech Republic was fully vaccinated. The difference is statistically significant (*p* < 0.0001, chi-square with Yates’ correction, OR = 8.167, 95% CI 7.438 to 8.966) and is illustrated in Figure 6. The results of this subchapter are further analyzed in the Discussion section.

#### 3.4.9. Type of COVID-19 Vaccine Received

Participants who stated that their COVID-19 vaccination had started or been completed were asked about the type of vaccine they received. Out of 2243 respondents, 1981 (88.3%) stated they received Comirnaty vaccine (Pfizer–BioNTech), 144 (6.4%) received the Spikevax vaccine (Moderna), 103 (4.6%) received the Vaxzevria vaccine (AstraZeneca, Oxford Biomedica Limited), 14 (0.6%) received the Janssen vaccine (Johnson & Johnson), and 1 (0.04%) did not know.

#### 3.4.10. How Was Vaccination Managed

Participants who stated that their COVID-19 vaccination had started or been completed were asked how their vaccination was managed. Out of 2250 respondents, 1449 (65.3%) stated they were vaccinated in a specialized vaccination center, 437 (19.7%) replied a regional chamber office managed their vaccination, 205 (9.2%) stated that their vaccination was managed by their employer, 60 (2.7%) were vaccinated by a general practitioner, 47 (2.1%) stated they were vaccinated in a hospital, and 22 (1.0%) provided answers classified as “Others”. This topic is further discussed in the Discussion section.

#### 3.4.11. A COVID-19 Vaccine Breakthrough Infection Rate

Of those who were fully vaccinated, nine admitted that they were infected with COVID-19 after the vaccination was completed. This represents a 0.42% rate of breakthrough infection. Data on the prevalence of COVID-19 among respondents are available in our previous publication [3].

## 4. Discussion

Dentists are in close contact with patients, and their work environment is associated with high droplet formation. Such conditions predispose dentists to be prone to droplet infections, including COVID-19. Despite this fact, there is insufficient information on the impact of COVID-19 on dentists. Reflecting this situation, the Czech Dental Chamber initiated an extensive survey divided into three chapters: COVID-19 Prevalence, Vaccination, and Epidemiology. Almost a quarter of the entire chamber was involved, resulting in one of the largest national surveys among dentists, both absolutely and relatively. A study on the prevalence of COVID-19 among Czech dentists was published in November 2021, vaccination data are part of this work, and a chapter on epidemiology will be processed early in 2022, making Czech dentists one of the best-described professional groups during the COVID-19 pandemic.

As no information on the vaccination process among Czech dentists was available before the start of the survey, we first decided to find out what approach they have to vaccination, including persons registered for vaccination, followed by a question about the reason for their decision. This procedure was chosen to also include those who had the willingness to be vaccinated but had not been vaccinated yet. The results revealed that almost all participants who showed an interest in vaccination did not delay it, but they were already in the process of vaccination. Their reasons were more than 90% professional, and only 25 respondents stated that they were motivated negatively, e.g., avoiding travel restrictions. A significant proportion of unvaccinated respondents stated medically relevant reasons for avoiding vaccination, such as laboratory-tested antibodies or medical contraindication. In summary, 89.5% of all participants were either registered for vaccination, partly or fully vaccinated, or had objective reasons to avoid vaccination. Some of the medically relevant reasons for avoiding vaccination, such as pregnancy and breastfeeding, are not considered medically relevant reasons to avoid vaccination today. However, in the first half of 2021, there was insufficient information on the safety of vaccines for these groups. The official recommendation for vaccination of pregnant and breastfeeding persons against COVID-19 was not issued in the Czech Republic until June 2021 [17]. Therefore, the classification of these reasons as “medically relevant” is correct for evaluating the approach to vaccination at the time of the survey, as it was consistent with the current state of knowledge.

Overall, 79.6% of all participants had completed vaccination against COVID-19. This result shows a very positive attitude of Czech dentists to vaccination as its rate is significantly higher than in the general Czech population. However, the comparison of the proportion of vaccinated persons among the members of the Czech Dental Chamber and the entire general population is limited by the different approaches to vaccination on the basis of profession and age. First, health professionals had government-guaranteed priority access to vaccination because of their profession. The general population’s access to vaccination was restricted due to a lack of vaccines in the first half of 2021. Second, dentists are university-educated people; hence, the study group consists of adults of different ages than the general population. To reflect on the potential bias resulting from the professional and age specificities of our study group, we decided to include COVID-19 vaccination statistics of the Czech general population over 25 years old for comparison. As all respondents admitted that their vaccination was completed by the end of June 2021, the first time point for comparison was set at 30 June 2021. On this date, ~40.7% of the general Czech population over the age of 25 years were fully vaccinated [18,19]. To eliminate the professional advantage of dentists in having access to vaccination, the second time point for comparison was set at 30 September 2021. At that time, vaccination had been available for a sufficiently long time without professional restrictions to all adults. On 30 September 2021, ~67.5% of the general population over the age of 25 years were fully vaccinated [18,19]. From this analysis, it is clear that even after the elimination of bias influences (occupational and age restriction), vaccination among dentists remains significantly higher than in the comparable general Czech population. The authors considered it important that the results of this study are comparable with other studies. Since vaccination data are generally presented mostly as a percentage of the general population, we decided to present the data in the same way and comment on the population specifics of this study here in the Discussion. We believe that the data provide the highest possible informativeness and comparability in this manner.

The more positive approach to vaccination against COVID-19 among Czech dentists could be due to their education type, as all Czech Republic dentists have graduated from medical universities. This is in accordance with the findings of Siegler et al., who observed higher willingness toward COVID-19 vaccination among persons with a bachelor’s or graduate degree than among persons with lower education [20]. Additionally, a work of Abedin et al. indicates that people with a higher level of education have a more positive approach to vaccination [21]. It is also interesting that although the previous study revealed that the prevalence of COVID-19 was higher among Czech dentists than in the general Czech population, their willingness to vaccinate was higher [3]. On the other hand, the willingness to be vaccinated can be higher in dentists who are also willing to participate in this kind of study. Additionally, some respondents could not disclose a refusal to vaccinate for professional reasons and avoid social disapproval. For more reliable data, it would be necessary to have the respondents’ medical records available for verification, which is impossible due to the confidentiality of this data.

The vaccination rate among respondents in age categories over 60 years was notably higher (by 10–15%) than in younger groups. Such a result may be due to the greater danger of COVID-19 for the elderly. The highest proportion of fully vaccinated people was among males aged 60–70, and the lowest among females aged 30–40. The ratio of vaccinated individuals was statistically higher among men. However, we do not consider this result epidemiologically significant as rather sociological. Compared to the entire study population, fewer people under the age of 30 and over 70 are represented among the study participants. These differences partially balance each other, as the vaccination rate was lower among people under 30 and higher among people over 70 in the general Czech population. However, this result can still be a source of bias.

From the presented data, it is evident that over 73% of respondents stated that their vaccination was completed by the end of March 2021, the rest by the end of June 2021. None of the respondents indicated that their vaccination was completed after June 2021, although the survey ended on 4 September 2021. These results suggest that there was an apparent attitude towards vaccination among respondents. Those who wanted to be vaccinated did so by a majority as soon as possible. Those who avoided vaccinations have consistently maintained this approach. In future population studies among Czech dentists, it will be interesting to see how this approach changes after introducing further measures, such as compulsory vaccination of health professionals. This will be addressed in the next phase of our research.

The most commonly accepted type of vaccine among respondents was Comirnaty (Pfizer). We believe that this was for two reasons. First, this vaccine has been presented as the most effective. Second, Comirnaty (Pfizer–BioNTech) was the majority vaccine given in vaccination centers, and most respondents stated that they were vaccinated there. This was probably due to vaccination management in the Czech Republic that was based mainly on large vaccination centers. In the first half of 2021, the situation with access to vaccination was organizationally complicated, even for eligible groups such as health professionals. Most members of the Czech Dental Chamber managed their vaccination by themselves via official registration, but some used the assistance of the chamber. The purpose of the vaccination management question was to find out which part of the chamber members used the standard registration process, identify the chamber’s role in the organization, and the facilities in which the dentists were vaccinated. Unfortunately, within those who were vaccinated with the assistance of the chamber, it was not possible to decipher in which facility the chamber provided them with vaccinations. Despite this limitation, we decided to present these data as it provides important information about where the dentists were vaccinated.

The comparison of this study with the studies of other authors is limited by the different methodology, population, or time window of the studies. To date, there is only a very limited number of studies focusing on vaccination among dentists, and most of them focus only on respondents’ willingness to get vaccinated and not on actual vaccination data. At the same time, these studies usually included smaller numbers of participants compared to ours. Nasr et al. conducted an online survey involving 529 Lebanese dentists between 15 and 22 February 2021 [11]. A total of 86% of participants were willing to receive or have already received a COVID-19 vaccine. Compared to our study, in which 83.9% of participants answered that they were either registered for vaccination or that vaccination had already started or was completed, this is a similar result. The authors also mention that the high level of vaccine acceptance is probably due to the high occupational risk of COVID-19 infection. This also corresponds with the results of our study, as 91.5% of respondents stated that the reasons for vaccination were primarily professional. According to an online survey performed by Zigron et al., the overall rate of acceptance for a COVID-19 vaccine among Israeli dental professionals during March-April 2020 was 85% [12]. A study by Papagiannis et al. among healthcare professionals involving 80 dentists in December 2020 showed a 78.5% acceptance level of the COVID-19 vaccine [13]. Another online survey performed among 2678 healthcare workers in France and French-speaking parts of Belgium and Canada in October and December 2020 showed a 71.6% high or moderate acceptance rate of the COVID-19 vaccine [14]. A COVID-19 vaccine acceptance survey among US healthcare workers performed in October and November 2020 showed that 36% of respondents were willing to take the vaccine as soon as it became available, 56% were not sure or would wait to review more data, and 8% did not plan to get the vaccine [15]. The data from these studies generally show similar results to our study. However, they identify only a possible approach to vaccination, not a result that reflects a real vaccination rate. Our work is one of the first to provide real data on healthcare workers vaccination and, as far as we know, the first to provide data on vaccination among a completely homogeneous population of dentists. The next phase of the research describing the impact of COVID-19 on Czech dentistry will be initiated in 2022, providing further comparable epidemiological and sociological data.

## 5. Conclusions

The study provided information on COVID-19 vaccination characteristics among the members of the Czech Dental Chamber during the pre-booster vaccination phase. The results revealed that the acceptance of COVID-19 vaccination and the COVID-19 vaccination rate were significantly higher among Czech dentists compared to the general Czech population.

## Figures and Tables

**Figure 1 vaccines-10-00428-f001:**
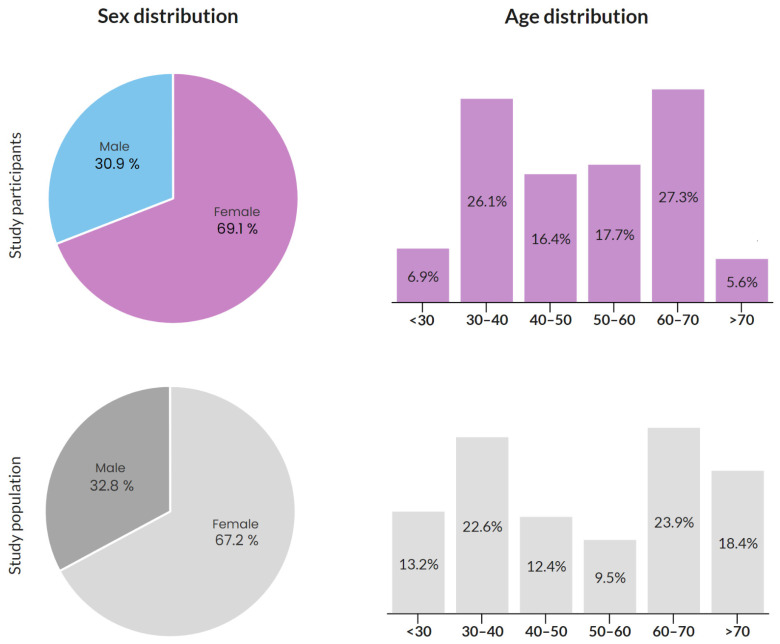
Sex and age distribution of the study participants and the study population.

**Figure 2 vaccines-10-00428-f002:**
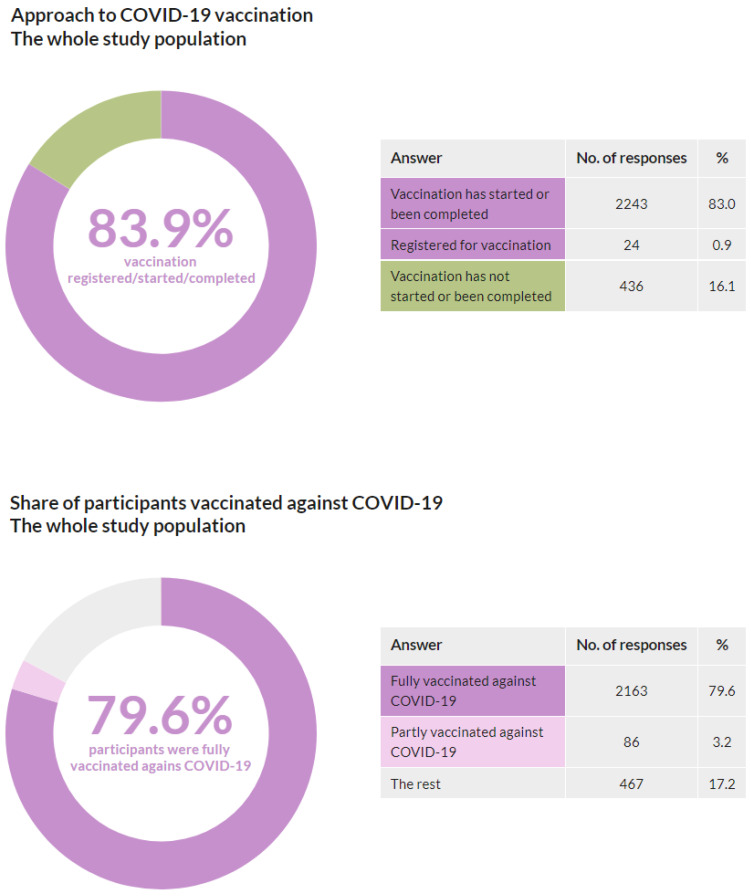
Approach to vaccination, the entire study population, and share of participants vaccinated against COVID-19.

**Figure 3 vaccines-10-00428-f003:**
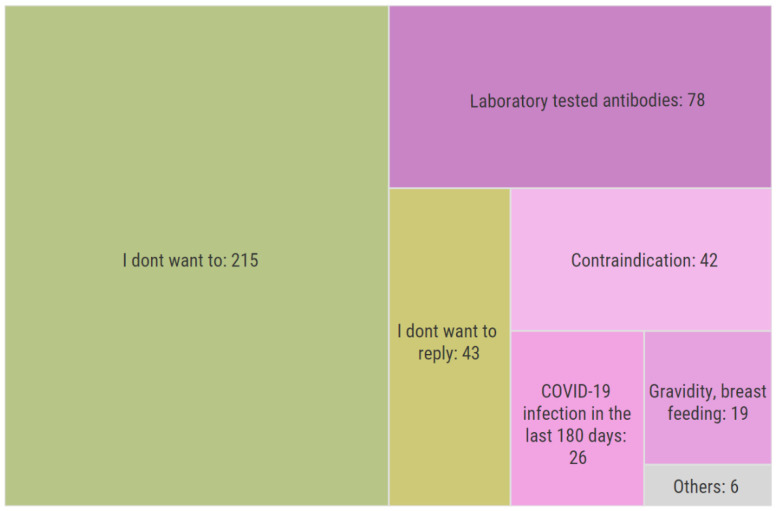
Reasons leading to avoiding COVID-19 vaccination.

**Figure 4 vaccines-10-00428-f004:**
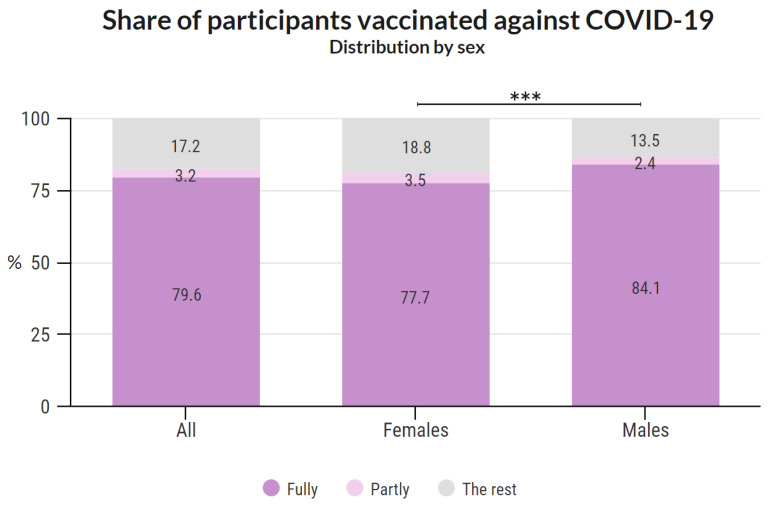
Share of participants vaccinated against COVID-19 based on sex. The difference in vaccination with at least one dose between males and females was statistically significant (*** indicates *p* = 0.001; chi-square with Yates’ correction; OR = 1.48, 95% CI 1.045 to 1.862).

**Figure 5 vaccines-10-00428-f005:**
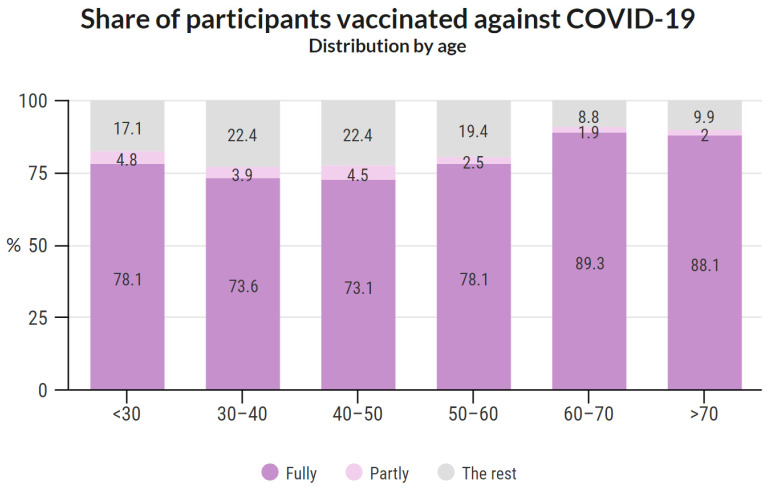
Share of participants vaccinated against COVID-19 based on age.

**Figure 6 vaccines-10-00428-f006:**
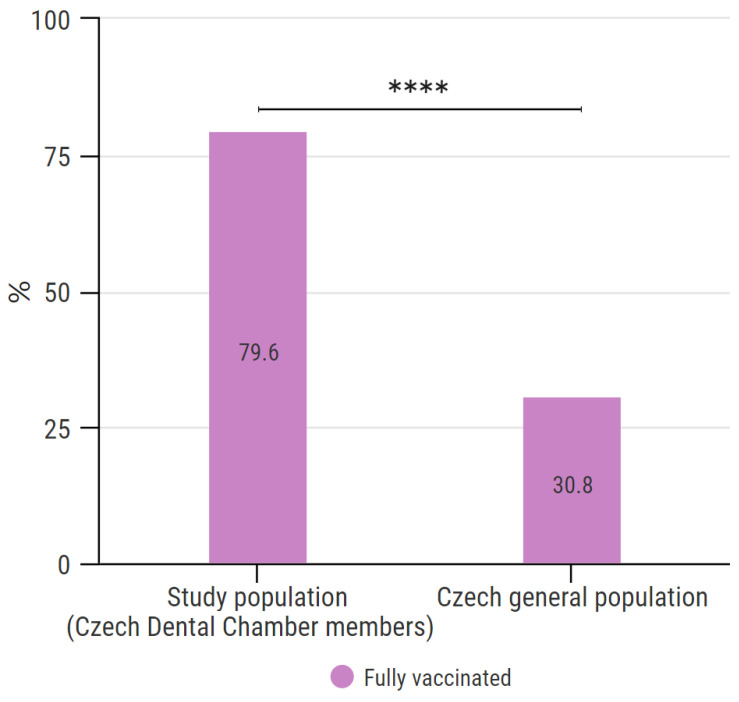
Comparison of full COVID-19 vaccination within the study population and Czech general population. **** indicates *p* < 0.0001, chi-square with Yates’ correction, OR = 8.167, 95% CI 7.438 to 8.966).

## Data Availability

The dataset is available on demand from the corresponding author.

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
