# Peer review of "COVID-19 Vaccination among Czech Dentists"

_vaccines, 2022, doi:10.3390/vaccines10030428_

Round 1

Reviewer 1 Report

This is an interesting manuscript addressing anti-COVID vaccination status among a specific group. Schmidt et al used data from an online survey assessing vaccination status among members of the Czech Dental Chamber. Overall, they evaluated 2713 responses and found a vaccination rate of 79.6% plus 9.9% already registered for it or with a medical reason not to do so. These figures were significantly higher than the general population. Breakthrough infection were self reported in 0.42% of vaccinated respondents.

The study provides some interesting data. However, I believe that it should be improved:

1- First, I feel that the work is too extensive and provide an excessive number of figures that made it difficult to read. I suggest that Table 1 should be provided as Supp material, Figure 1 do not bring any information outside of what is already written in the manuscript, Figure 2 and 3 can be merged, Figure 5 can be removed, Figure 4 and 7 can be merged. Figure 10 also summarize figure 8 and 9. Figure 11 is also not informative. The month when recommendations for vaccination were made and the same was logistically possible was not the manuscript objective.

Figures 13 and 14 also are not in the scope of the manuscript and can also be removed.

2- Although authors referred to a larger study addressing the impact of COVID 19 in dentists, including the rate of infection, that has previously been published, no such data is provided anywhere in this manuscript. I think authors should provide a short description of what was found. The same would be particularly interesting as the rate of infection was actually lower than the rate in the general population. This numbers could have an impact on the vaccination rate but that did not happen, which is interesting.

3- According to the study design, respondents were free to do so. Accordingly possible bias was introduced. Willingness to be vaccinated can be much higher in those dentists who are also willing to participate in this kind of studies. Authors should at least provide a description of age and gender of participants and of the whole study population to allow identification of significant differences. This should also be disclosed in the study limitations.

4- Moreover, respondents may fail to disclose a refusal to vaccinate, both for professional reasons and to avoid social disapproval. Authors should comment on this.

5- Comparison between genders and age groups in the rate of vaccination (figures 8 and 9) do not seem to considerer medical reasons to avoid vaccination. As the authors considerer those valid, this comparison should be made after excluding patients that cannot be vaccinated.

6- In discussion, lines 277-295, authors compared the rate of vaccination among dentists to the general population. However, as all dentists are adults, the control group should be the adult population, at least over 21.

Reviewer 2 Report

 COVID-19 Vaccination among Czech Dentists

This work describes and evaluates vaccination against COVID-19 among members of the Czech Dental Chamber during the pre-booster vaccination phase. A cross-sectional online survey was conducted between June 23 and September 4, 2021, among 2716 participants .

The plus point of this manuscript is its coherence. The weak points are the lack of rigour in term of content and presentation.

I listed my comments below:

Ambiguity:

Our team was the first to publish a study on the national prevalence (of what ?) of dentists from the beginning of the pandemic  to the end of the first half of 2021 [12].

ad hoc survey (?)

and its participants did not have patient status (?)

Unverified info:

Coronavirus disease COVID-19 is one of the major health concerns since it was first 28 detected in Wuhan City, Hubei Province, China (?), in December 2019
Ref: https://www.ncbi.nlm.nih.gov/pmc/articles/PMC7982270/

No curative therapy (?)  for COVID-19 has been discovered so far, and preventive 38 measures are the only tool to mitigate the pandemic. Ref: https://emedicine.medscape.com/article/2500116-overview

The results of the Prevalence chapter were published in November 2021, the data of the Vaccination chapter are presented in this work, and the Epidemiology chapter will be published in 2022

Comment: it seems a salami slicing practice. Salami slicing refers to the practice of partitioning a large study that could have been reported in a single research article into smaller published articles. A set of papers are referred to as salami publications when more than one paper covers the same population, methods, and research question.

Conclusions

Of the 2716 members of the Czech Dental Chamber who participated in this study 407 from June 23, 2021 to September 4, 2021, 89.5% stated that they were registered for vac-408 cination against COVID-19, their vaccination had started or been completed, or had a 409 medically relevant reason to avoid vaccination. A total of 79.6% of respondents stated that 410 they were fully vaccinated, most of them with the Comirnaty (Pfizer–BioNTech) vaccine 411 (88.3%). The main reasons for the vaccination were professional (91.5%). The share of fully 412 vaccinated participants was significantly higher compared to the Czech general popula-413 tion (30.8%). A COVID-19 vaccine breakthrough infection was 0.42%. The study shows 414 that both the willingness to vaccinate and the proportion of fully vaccinated people among 415 Czech dentists are high, and only about 10% of them refused vaccination on the basis of 416 reasons classified as not medically relevant.

Comment: this is merely repeating the presented info. Please summarise the findings.

Discussion:

The rate of a COVID-19 vaccine breakthrough infection was low (0.42%). This result  is in accordance with Bergwerk et al. who reported on COVID-19 breakthrough infections in vaccinated health care workers in Israel to be 0.4%

Comment: What is the relevance of including this.

  1. Study Limitations

The comparison of the proportion of vaccinated persons among the members of the 277 Czech Dental Chamber and the whole general population is limited by the different ap-278 proaches to vaccination on the basis of profession and age.

The second limitation concerns vaccination management. In the first half of 2021, the 296 situation with access to vaccination was organizationally complicated, even for eligible 297 groups such as health professionals.

The third limitation is the classification of medically relevant reasons for avoiding 308 vaccination. Some of these reasons, such as pregnancy and breastfeeding, are not consid-309 ered medically relevant reasons to avoid vaccination today

Comment: these are the inherent shortcomings of the system and not the limitation exerted of the operation of this study

Table 1. Questions and their classification

Figure 1. Response rate. The number of participants in this study is 2716, which makes 27.4% re-146 sponse rate within 9922 e-mails sent and 24.3% participation rate within all Czech Dental Chamber 147 members.

Figure 2. Study participants’ sex distribution

Figure 3. Age distribution of the study participants.

Figure 4. Approach to vaccination, the whole study population

Figure 5. Reasons leading to COVID-19 vaccination

Figure 6. Reasons leading to avoiding COVID-19 vaccination

Figure 7. Share of participants vaccinated against COVID-19

Figure 8. Share of participants vaccinated against COVID-19 based on sex. The difference in vac-214 cination with at least one dose between males and females was statistically significant (p = 0.001; 215 chi-square with Yates' correction; OR = 1.48, 95% CI 1.045 to 1.862).

Figure 9. Share of participants vaccinated against COVID-19 based on age

Figure 10. Share of participants vaccinated against COVID-19 based on age and sex. Values are 224 given as percentages

Figure 11. The month when full vaccination was achieved

Figure 12. Comparison of full COVID-19 vaccination within the study population and Czech general 245 population.

Figure 13. Type of COVID-19 vaccine received.

Figure 14. Vaccination management

Comments: Table 1 and 14 figures: there are a lot of duplication/redundancy of presented info and demographic of the respondents should be included instead.

Reviewer 3 Report

Dear Editor and Authors,

It was my pleasure to review this manuscript titled “COVID-19 Vaccination among Czech Dentists” by Dr. Schmidt and his colleagues from the Czech Republic. In this questionnaire analysis the authors have investigated the acceptance of Covid-19 pre-booster (2 doses) vaccination amongst their dentist colleagues. To ascertain this, the authors as part of a mandated chamber committee, conducted an online survey amongst members of the Czech Dental Chamber. They were able to obtain responses from 2716 dentists which represents almost a quarter (24,3%) of all registered dentists, which is a significant sample size and more that covers the sample size calculated by the authors, however sadly it is small if one considers that the chamber has almost 11000 members and 9922 of those were conducted!

This is a well written manuscript and a pretty simple epidemiological survey. The language is good with only minor mistakes which can be polished by a native English speaker or a professional editing service.

The introduction is a bit long and verbose analyzing the whole history of the Covid-19 pandemic. Given that all this is well known to the medical community I believe we can safely omit it. I would therefore suggest deleting the first 2 paragraphs and starting the introduction from the third one onwards.

The methodology of the study is straightforward and uncomplicated and the results are well presented with nice, clear illustrations and tables. I found the results not unexpected amongst medical professionals. One limitation, I might add to the methodology of the study is the fact that dentists which do not wish to get vaccinated and are “suspicious” of the anonymity of the survey or slight extreme in their views might not opt to participate in the survey!

One more comment has to do with the limitations of the study which appears to contain parts (for example discussion about children) that more aptly pertain to the discussion section. This is something that even the authors have identified in their discussion section (line 344).

One final comment I have is that I did not see that the study had secured some form of ethical approval from the chamber or some other institution/university.

In conclusion, this is an interesting epidemiological presentation of the vaccination prevalence and views of a specific health care group (dentists) in the Czech Republic. It is well conducted and presents interesting findings and therefore I would be inclined to recommend its publication after some minor editing from the authors. I wish all well.

Round 2

Reviewer 1 Report

The authors have significantly improved their manuscript. I have no further comments

Reviewer 2 Report

All comments have been addressed. 

This manuscript is a resubmission of an earlier submission. The following is a list of the peer review reports and author responses from that submission.